# Spatial Distribution of Drug-Resistant *Mycobacterium tuberculosis* Infections in Rural Eastern Cape Province of South Africa

**DOI:** 10.3390/pathogens12030475

**Published:** 2023-03-17

**Authors:** Lindiwe M. Faye, Mojisola C. Hosu, Sandeep Vasaikar, Anzaan Dippenaar, Selien Oostvogels, Rob M. Warren, Teke Apalata

**Affiliations:** 1Department of Laboratory Medicine and Pathology, Walter Sisulu University and National Health Laboratory Services (NHLS), Private Bag X5117, Mthatha 5099, South Africa; 2Family Medicine and Population Health (FAMPOP), Faculty of Medicine and Health Sciences, University of Antwerp, BE-2000 Antwerp, Belgium; 3DSI-NRF Centre of Excellence for Biomedical Tuberculosis Research, SAMRC for Tuberculosis Research, Division of Molecular Biology and Human Genetics, Faculty of Medicine and Health Sciences, Stellenbosch University, Cape Town 7505, South Africa

**Keywords:** tuberculosis, spatial analysis, mutations, spoligotypes, heteroresistance

## Abstract

Tuberculosis (TB), an infectious airborne disease caused by *Mycobacterium tuberculosis* (Mtb), is a serious public health threat reported as the leading cause of morbidity and mortality worldwide. South Africa is a high-TB-burden country with TB being the highest infectious disease killer. This study investigated the distribution of Mtb mutations and spoligotypes in rural Eastern Cape Province. The Mtb isolates included were 1157 from DR-TB patients and analysed by LPA followed by spoligotyping of 441 isolates. The distribution of mutations and spoligotypes was done by spatial analysis. The *rpoB* gene had the highest number of mutations. The distribution of *rpoB* and *katG* mutations was more prevalent in four healthcare facilities, *inhA* mutations were more prevalent in three healthcare facilities, and heteroresistant isolates were more prevalent in five healthcare facilities. The Mtb was genetically diverse with Beijing more prevalent and largely distributed. Spatial analysis and mapping of gene mutations and spoligotypes revealed a better picture of distribution.

## 1. Introduction

Tuberculosis (TB), a chronic inflammatory infectious disease caused by *Mycobacterium tuberculosis* (Mtb), is a serious public health threat and is reported as the leading cause of morbidity and mortality worldwide. It is easily transmitted from one person to another by airborne droplet nuclei [1,2]. In 2021, the most current global TB data reported the incidence of TB to be 10.6 million new cases with 1.6 million attributable fatalities globally [3]. The distribution of TB differs geographically both within counties and on different continents of the world. Africa accounts for 29% and 34% of all TB cases and deaths, respectively worldwide with the highest recorded incidence rate of 275 cases per 100,000 people [4,5].

South Africa is a high TB burden country with TB being the highest infectious disease killer. According to the global burden of disease study, TB is the fifth leading cause of years of life lost (YLL) and disability-adjusted life years (DALY) in the country [6]. With an estimated population of 60.6 million by the end of June 2022 [7], South Africa (SA) shares borders with 6 other nations, including Botswana, Lesotho, Namibia, Mozambique, Swaziland, and Zimbabwe, where TB is also endemic. In 2019, the TB incidence in South Africa was estimated to be 615 cases per 100,000 population, ranging from 427 to 835 cases per 100,000 and with estimated 360,000 people who developed TB [8]. SA had the second-highest absolute number of notified rifampicin (RIF)-resistant and multidrug-resistant (MDR) cases globally with 18,734 cases reported in 2015 [9]. Eastern Cape is one of the three provinces in South Africa that have the highest TB incidence rates [1].

A better understanding of local geographic heterogeneity in routinely identified TB cases and the correlation of that heterogeneity with the location of undiagnosed prevalent cases may, therefore, be useful in directing active case-finding interventions to high-risk areas [10]. Furthermore, there is a need for accurate and early detection of drug-resistant TB (DR-TB) for minimizing the development of drug resistance, effective patient care, and preventing the spread of DR strains [11].

Our knowledge of the epidemiology of TB on a local and global level has been substantially improved by the development and use of genotyping methods for Mtb; likewise, with the help of geospatial analytical technologies, the understanding of public health issues can be enhanced [12,13]. Spoligotyping combined with geospatial analytical methods, such as geographic information systems (GIS) and directly observed treatment short-course (DOTS) strategy, can help assess the transmission of Mtb strains and are promising essential tools for helping to understand the distribution of drug-resistant strains as well as the drivers of drug resistance [14]. Previous spatial studies have used GIS, whole genome sequencing (WGS), and spatial statistics to identify transmission hotspot areas with elevated risk for prioritisation of control and intervention measures [15,16,17]. These spatial studies have also shown that MDR-TB is clustered in specific geographical areas associated with location, socio-economic status, and population density [4]. Even though there is an increase in the number of studies that use geospatial analytical methods to understand TB and other public health problems [18,19,20], however, the transmission dynamics of prevalent Mtb strains in rural Eastern Cape are not well understood. 

Understanding such spatial variations in TB prevalence is crucial for improved targeting of interventions and resources for the prevention and management of TB in a particular area. The geographical distribution of MDR-TB in high TB burden settings is very important for the effective control of TB epidemics. This can inform a basis for understanding DR gene migrations within populations since the frequency of mutations varies geographically. In this study, we sought to investigate the spatial patterns of Mtb in order to determine the transmission patterns and mixed-strain infections. We present the first spatial analysis of DR-TB and mutations associated with RIF and isoniazid (INH) causing heteroresistance and spoligotype distributions of Mtb in healthcare facilities (HCFs) of Mthatha and surrounding areas.

## 2. Methods

### 2.1. Study Design and Setting

This was an ecological study design conducted in 4 districts, namely Oliver Reginald Tambo (O. R. Tambo), Alfred Nzo, Amatole, and Chris Hani, and 1 metropolitan municipality (Buffalo City), with 101 healthcare facilities in rural Eastern Cape Province (ECP), South Africa; the distribution of the healthcare facilities is as shown (Figure 1).

Eastern Cape province is the second largest province in the country and serves a population of 7,130,480. O. R. Tambo district is 1 of the 7 districts of the ECP located on the coastline. The seat of O. R. Tambo is in Mthatha. The population is about 1,676,463. Five local municipalities, namely King Sabata Dalindyebo, Nyandeni, Mhlontlo, Port St. Johns, and Ingquza, form O. R. Tambo district municipality.

### 2.2. Study Population and Sampling Strategy

The study population included all DR-TB cases registered and living in the catchment area of the five municipalities’ TB treatment centres and Nelson Mandela Academic Hospital National Health Laboratory Services (NHLS) catchment areas, registered between the years 2018 and 2020. 

### 2.3. Data Collection and Isolates Identification

Sputum samples were collected from consecutive clinically diagnosed pulmonary TB patients reporting to 101 selected health facilities within the district municipalities named above. The samples were submitted to NHLS TB Laboratory for diagnostic testing. A convenience sample of 1157 Mtb isolates was selected to include: (1) INH monoresistance (IMR-TB) defined as resistance to a single first-line drug, isoniazid, (2) RIF monoresistance (RMR-TB) defined as resistance to a single first-line drug, rifampicin, (3) MDR-TB defined as resistant to at least isoniazid and rifampin, the two most potent TB drugs, and (4) heteroresistance defined by the coexistence of susceptible and resistant organisms to anti-tuberculosis drugs in the same patient (defined by the hybridisation to both the wild type (WT) and mutant (MUT) probes on the MTBDRplus assay). The information of patients on the laboratory requisition form was recorded for spatial analysis in order to determine the distribution of mutations in *rpoB*, *katG*, and *inhA* genes and their spoligotypes among isolates. Mutations were determined by the presence of the binding of amplicons to probes targeting the most commonly occurring mutation probes or inferred by the lack of hybridization (i.e., lack of binding) of the amplicons to the corresponding WT probes. Isolates were collected over a 3-year period and investigated using first-line probe assay (LPA).

### 2.4. Isolates Analysis 

The GenoType MTBDRplus VER 2.0 is a DNA-strip based in vitro assay for identifying the Mtb complex (MTBC) and its resistance to RIF and INH from smear-positive pulmonary sputum samples and positive culture samples. DNA was extracted using Genolyse^®^ kit (Hain Life Science GmbH, Nehren, Germany). The extracted DNA was processed by the LPA using DRplus [21] to detect MTBC and RIF and/or INH resistance according to the manufacturer’s instructions. Out of 1157 isolates, 441 were conveniently selected to include IMR-TB, RMR-TB, and MDR-TB and heteroresistant isolates for spoligotyping to determine genotypes of Mtb isolates circulating in Mthatha and surrounding areas. Spoligotyping was performed using microbeads from TB-SPOL Kit (Beamedex^®^, Orsay, France), and the fluorescence intensity was measured using Luminex 200^®^ (Austin, TX, USA). Generated patterns were assigned to families using the SITVIT2 international database of the Pasteur Institute of Guadeloupe and compared [22]. 

The LPA and spoligotype results of Mtb isolates were analysed for distribution of the mutations and spoligotypes using the QGIS 3.14 software. LPA score and banding patterns were used to determine the type of DR-TB and heteroresistance. Clinics within hospitals with the same coordinates were merged in the analysis. We assessed the distribution of mutations in *rpoB, katG*, and *inhA*, as well as the distribution of heteroresistant strains and spoligotypes.

## 3. Results

### Distribution of Mutations, Lineages of Isolates

A total of 1157 DR-TB and heteroresistant clinical isolates were isolated from the different healthcare facilities. 

RMR was represented by the LPA score of *rpoB* MUT/*katG* WT/*inhA* WT, while IMR was represented by *rpoB* WT/*katG* WT/*inhA* MUT or *rpoB* WT/*katG* MUT/*inhA* WT and *rpoB* WT/*katG* MUT/*inhA* MUT LPA score. The MDR-TB strains were represented by LPA score of *rpoB* MUT/*katG* WT/i*nhA* MUT or *rpoB* MUT/*katG* MUT/*inhA* WT and *rpoB* MUT/*katG* MUT/*inhA* MUT. Heteroresistant strains were represented by LPA score where there were both mutation and wildtype bands on any of the three genes (*rpoB*, *katG*, and *inhA*). 

The number of isolates from 6 HCFs that had a higher proportion of gene mutations and spoligotypes was HCF1 (78/1157), HCF 2 (94/1157), HCF 3 (114/1157), HCF 4 (83/1157), HCF 5 (73/1157), and HCF 6 (55/1157). All three (*rpoB*, *katG* and *inhA*) genes under investigation had one or more mutations from each isolate. The total number of mutations that occurred in the *rpoB* gene was 761, representing the highest number of mutations, followed by the *katG* gene with 683 mutations, while the *inhA* gene accounted for 286 mutations. The distribution of these *rpoB* mutations is shown in Figure 2. A total of 4 healthcare facilities had more gene mutations, namely HCF 3 (n = 83), HCF 2 (n = 77), HCF 4 (n = 60), and HCF 1 (n = 53).

*katG* gene mutations are shown below (Figure 3). A total of 4 healthcare facilities had more mutations namely HCF 4, n =36, HCF 2, n = 42, HCF 1, n = 43, and HCF 3, n = 56.

There were 3 healthcare facilities that had the highest number of *inhA* gene mutations, HCF 5, HCF 4, and HCF 2, with 28, 32, and 37 mutations, respectively, indicated in Figure 4. 

The total number of heteroresistant isolates was 207, which was 17.9% of the total number of isolates. These isolates had one or more mutations in different genes. The following clinics had the most mutations in heteroresistant isolates, HCF 1, HCF 4, HCF 2, HCF 5, and HCF 3, with 78, 83, 93, 94, and 114 mutations, respectively (Figure 5). 

The prevalent gene mutations in HCF within three municipalities were captured in the table below (Table 1).

Based on the 441 spoligotyping results, 8 families were identified. Of 441 isolates queried for the lineage assignment, 59 (81.9%) were classified into the previously known lineages, and 13 (18.1%) were not assigned to any known lineages (Table 2). The Beijing family was the predominant group, representing 42.0% (n = 185) of all isolates, followed by the LAM family, 18.8% (n = 83), X family, 10.9% (n = 48), T family, 7.7% (n = 34), S family, 7.0% (n = 31), EAI family, 3.6% (n = 16), H, 1.4% (n = 6), and CAS family, 1.1% (n = 5) (Figure 6). Only 4 (0.9%) isolates showed unknown patterns that were not assigned to any known major lineages in the SITVIT2 database, and 27 (6.1%) had no results. 

Although spoligotyped isolates were received from many HCFs of the study setting, HCFs with greater than n = 30 isolates were from 2 municipalities, namely O. R. Tambo (HCF 2, HCF 3, HCF 5 and HCF 6) and Buffalo City (HCF 4) (Figure 7).

Our results showed that from the 441 clinical isolates that were spoligotyped, the Beijing family strains accounted for 42.0% (185/441) of all the strains circulating in Mthatha and surrounding areas. The healthcare facilities with the most Beijing family strains were HCF 2 (n = 23), HCF 3 (n = 18), HCF 5 (n = 15), and HCF 6 (n = 13) (Figure 8).

The LAM family being the second most prevalent was mostly identified in two HCFs (HCF 4 and HCF 5) (Figure 9).

## 4. Discussion

The substantial gap in the detection and treatment of MDR-TB/RMR-TB means that most patients are missed. Hence, identifying geographical areas with a high incidence of disease and adopting active case findings in such areas could help to reduce the detection gap among DR-TB patients. By highlighting high-burden regions with poor public health initiatives and low case detection rates, knowledge of the spatial distribution of tuberculosis could help with control and prevention efforts. This identification enables decision-makers to undertake targeted measures for the prevention and management of MDR-TB hotspots in order to stop the spread of the disease [4,23,24]. This may be crucial in areas with limited resources and nations with high MDR-TB burden [4]. This is the first epidemiological study in the Eastern Cape combining GIS analysis with molecular-based methods to describe the distribution of DR-TB by mapping the distribution of DR gene mutations and genotypes.

Strategic measures in combating the spread of DR-TB include active surveillance, screening for DR-TB, early isolation, and management of confirmed cases [25]. The first step in easing the burden of DR-TB is to comprehend the geographical distribution of the disease and identify regions with the greatest prevalence of notified cases [23]. Drug resistance in Mtb is singularly mediated by chromosomal mutations [25]. Our study detected mutations in *rpoB, katG*, and *inhA* from the four municipalities. Most of the mutations are concentrated in the O. R. Tambo district municipality. This may be because it is more populous than the other municipalities. HCF 5, HCF 2, and HCF 4 had the highest number of mutations in the *inhA* gene (Figure 4); this means INH can still be used but in high doses for the treatment of TB in patients from the catchment areas of the clinic, however, due to development of mutations in genes during treatment, treatment management is needed. These HCFs are scattered in the municipalities, and they are not equidistant to one another. Our study setting being burdened with TB makes it easy to exchange resistant strains which increases mutation rates. The study setting has more of the Beijing family known to be more transmissible than other families and more prevalent in Western Cape Province, which is a neighboring province. Published evidence reviewing the frequency and distribution of gene mutations in Africa reported that out of the 22 gene mutations reported from 25 countries in Africa, *rpoB* ranked the highest. It further reported that mutations of *rpoB*, *katG*, *gyrA*, and *inhA* are documented in almost all regions of Africa, which is an indication of the widespread rifampicin- and isoniazid-resistant TB throughout the entire continent. Hence, TB treatment in Africa using rifampicin, isoniazid, fluoroquinolone, and bedaquiline should be handled with caution [26].

The coexistence of both drug-susceptible and drug-resistant bacteria within the same patient was observed at 17.9% of the total isolates of the study with 5 HCF that had the most genes; the *rpoB* gene (RIF resistance), the *katG* gene (high-level INH resistance), and the promoter region of the *inhA* gene (low-level INH resistance) with heteroresistance. This may contribute to the difficulty in treating tuberculosis, as it is the precursor to full resistance [27,28,29]. Heteroresistance is a result of mixed infection, with two or more Mtb distinct strains in the same patient or the presence of different subpopulations caused by the microevolution of the single strain within the host [30]. This may endanger the effective treatment of patients with both RIF and INH, thereby leading to the development of anti-TB drug resistance in the study area, which underscores surveillance of heteroresistance from patients prior to and after treatment. The changing drug resistance patterns detected in patients with tuberculosis also confirmed the possibility that heteroresistance can persist over a long period [31]. Studies on heteroresistance have reported a prevalence ranging from 20% to 57% [32,33,34], which was higher than our study but observed to be increasing each year. Most of the studies focused on samples taken before treatment was initiated, presumably to show that the presence of heteroresistance should be considered in formulating treatment regimens. In our study, the samples were collected during the treatment period, which means that heteroresistance can be present even if the patient is on treatment and agrees with van Rie et al. [31] who reported the persistence of heteroresistance over a long period. The detection of heteroresistance is vital to preventing the selection of drug resistance during antibiotic treatment [35].

Genomic analysis of Mtb clinical isolates worldwide has revealed differences in the geographical distributions and apparent host preference of distinct phylogenetic lineages [36]. According to [37,38,39], genomic analysis helps to identify clinical features of predominant or emerging genotypes and is important from a public health perspective because they describe epidemiological associations with outbreaks and transmission routes. Studies that are investigating the relationship among Mtb across different geographical areas are impacting positively the programs set to end TB because they help to understand the transmission of TB [40].

Of 441 isolates spoligotyped, 437 revealed distinct spoligotype patterns. Patterns of 410 (93.1%) isolates matched a pre-existing SIT in the SITVIT2 database, while 27 unique patterns (6.1%) were not in the database. The Mtb population in this study area was genetically diverse with the Beijing lineage and its members, which is regarded as a successful clone of Mtb associated with drug resistance in some parts of the world [41]. A comparison of lineages among all clinics/hospitals in this study shows that the Beijing family was the commonest genotype found in all the hospitals/clinics, with HCF 2, HCF 3, HCF 5, and HCF 6 having a higher proportion of Beijing isolates. The Beijing family is known to be more transmissible than other families and more prevalent in Eastern Cape and Western Cape provinces, which are neighbors [42,43]; this confirms its high prevalence in this study (42%). This family has been detected in studies reported from other parts of South Africa, including Limpopo, the Western Cape, and Mpumalanga [43,44]. Nelson et al. [45] reported that human movement between rural and urban areas in search of employment in South Africa is common and serves as a bridge for transporting pathogens across long distances. The LAM family was the second most prevalent. This strain is widely distributed in KwaZulu Natal, which is a neighboring province to the Eastern Cape [46]. 

Table 3 compares the distribution of the lineage of Mtb from our study with that in other studies, including other provinces in South Africa and sub-Saharan Africa. The Beijing family belonging to Lineage 2 is the most prevalent in our study, which is the same with the Western Cape, Gauteng, and North-West, but a different outlook is portrayed in countries such as Zambia and Botswana with LAM family predominating. Major lineages, L1 to L7, have been identified from analyses of Mtb strains worldwide [47,48,49], but recently this has been updated to include Lineages 8 and 9 [49]. Lineage 2 includes strains, the majority of which are members of the so-called Beijing family. More than a quarter of the world’s tuberculosis epidemic is attributable to the Beijing family, which is the most prolific genotype of Mtb. The widespread proliferation of this strain family in recent decades, its propensity to spread disease, association with drug resistance, treatment failure, early relapse, recurrence, fever during early therapy, and increased risk of transmission chains globally have all garnered considerable attention, according to reports from several clinical trials. Evidence from both experimental and clinical data points to Beijing strains’ hypervirulent phenotype and increased mutation rate when compared to other strains [47,49,50].

Mtb genotypes differ amongst populations and are strongly influenced by geography. In terms of host immune response modulation, transmissibility, and disease severity, different Mtb lineages frequently exhibit distinct traits and virulence profiles. A better understanding of phenotypic variations caused by the genetic diversity of Mtb strains is important when attempting to improve TB control measures [49]. Previous research has revealed that immune responses are significantly variable amongst genetically diverse Mtb strains. Lineage 2 Mtb strains are the most virulent and were shown to only elicit a weak immune response in mice. Evidence revealed that patients who were infected with Lineage 2 strains were more likely to die of TB compared to patients infected with other strains. Investigating the pathogenicity of distinct lineages of Mtb is, therefore, crucial [50].

## 5. Conclusions

The key control for DR-TB is to interrupt its transmission; this was done by identifying hotspots of gene mutations and lineages, especially those that are drivers of DR-TB transmission and mixed infections. The identification of areas where DR-TB is concentrated could assist policy makers to implement targeted interventions aimed at the prevention and management of TB transmission. This is particularly important in resource-limited settings and high DR-TB burden areas, such as the rural areas of ECP. Targeted interventions to the rural community may be necessary as these areas find it impossible to provide DR-TB services across the communities as the diagnosis and treatment of DR-TB are challenged by factors such as poverty and co-infection with HIV.

## Figures and Tables

**Figure 1 pathogens-12-00475-f001:**
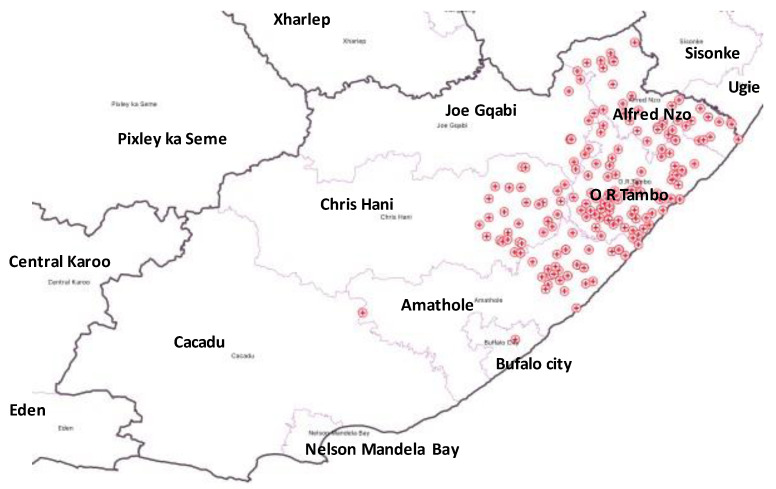
Distribution of healthcare facilities in the study area.

**Figure 2 pathogens-12-00475-f002:**
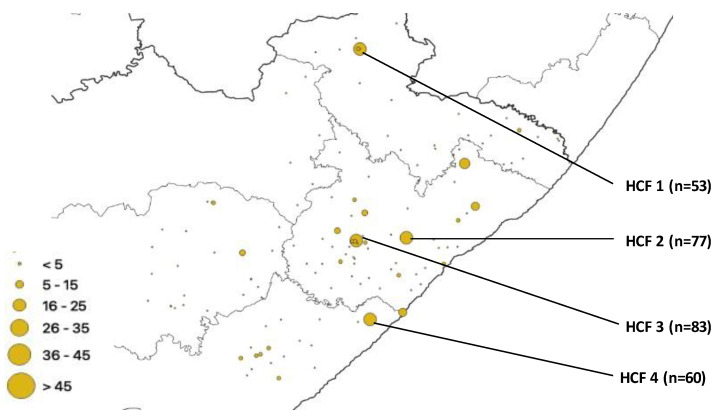
*rpoB* mutation distribution.

**Figure 3 pathogens-12-00475-f003:**
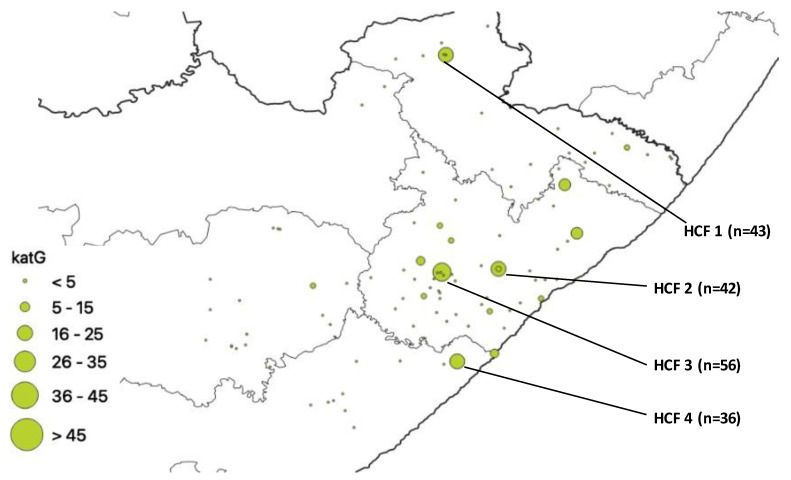
*katG* mutation distribution.

**Figure 4 pathogens-12-00475-f004:**
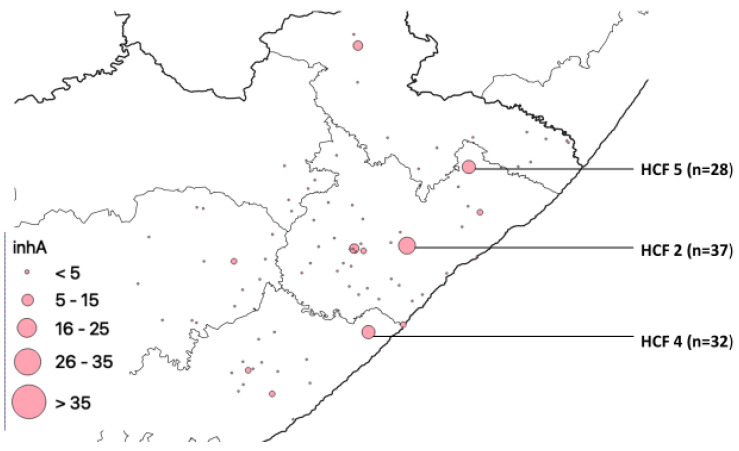
*inhA* mutation distribution.

**Figure 5 pathogens-12-00475-f005:**
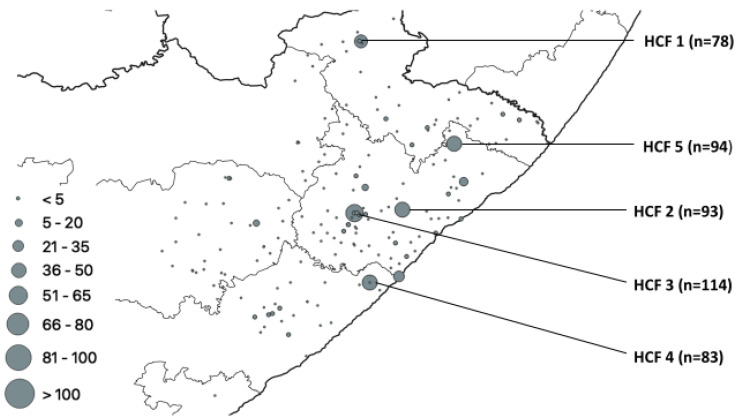
Geospatial distribution of heteroresistance in rural ECP.

**Figure 6 pathogens-12-00475-f006:**
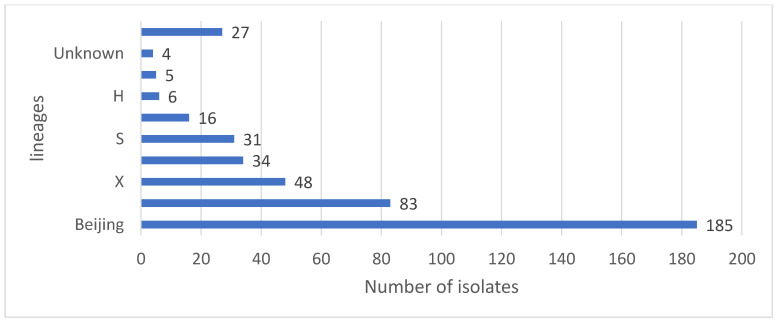
Distribution of Mtb lineages/families. (LAM: Latin American; EAI: East-African Indian; Delhi/CAS: Delhi/Central Asian, H: Haarlem).

**Figure 7 pathogens-12-00475-f007:**
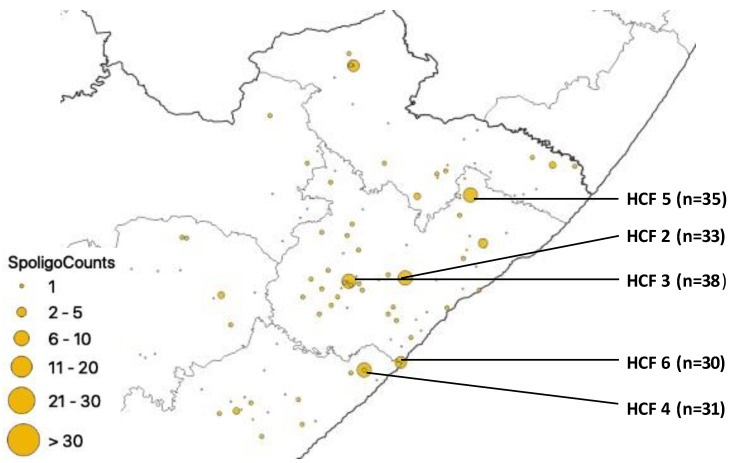
Geospatial distribution of spoligotyped isolates in healthcare facilities.

**Figure 8 pathogens-12-00475-f008:**
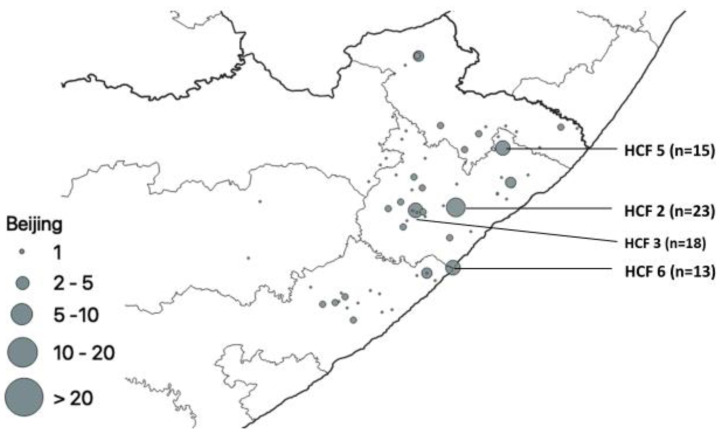
Geospatial distribution of Beijing family.

**Figure 9 pathogens-12-00475-f009:**
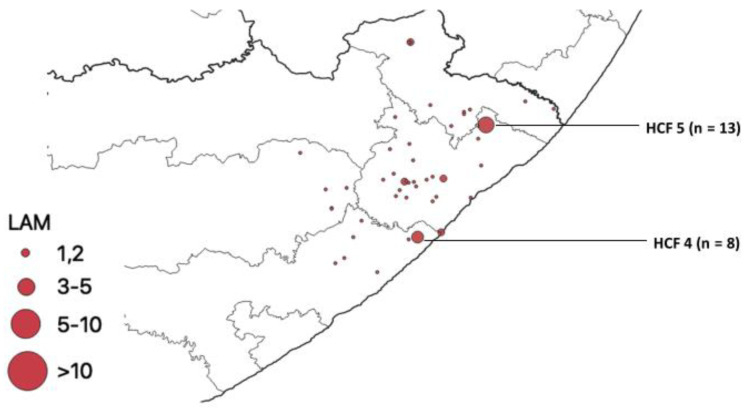
Geospatial distribution of LAM family in healthcare facilities in rural ECP.

**Table 1 pathogens-12-00475-t001:** Distribution of prevalent gene mutations in HCF in rural ECP.

HCF	Municipality	*rpoB*	*katG*	*inhA*	Heteroresistant Genes
1	Alfred Nzo	53	43	0	78
2	O. R. Tambo	77	42	37	93
3	O. R. Tambo	83	56	0	114
4	Amathole	60	36	32	83
5	O. R. Tambo	0	0	28	94

**Table 2 pathogens-12-00475-t002:** Spoligotyping Patterns.

Lineage	No of Isolates	Sublineage	No of Isolates	SIT	Spoligo Pattern	No of Isolates
Beijing	185			1	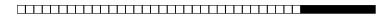	185
LAM	83	LAM3	62	33	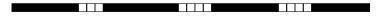	36
				719	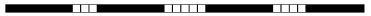	17
				ORPHAN	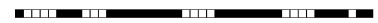	2
				4	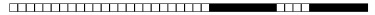	1
				130	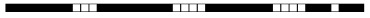	1
				376	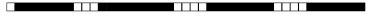	1
				2014	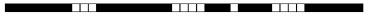	1
				2284	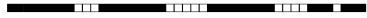	1
				2302	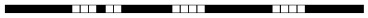	1
				not in SITVIT	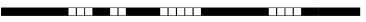	1
		LAM4	15	60	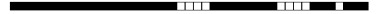	15
		LAM5	2	93	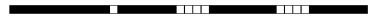	1
				136	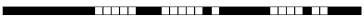	1
		LAM9	2	42	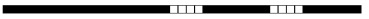	2
		LAM11-ZWE	2	811	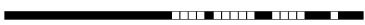	1
				1873	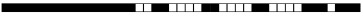	1
X	48	X1	18	2022	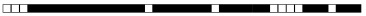	10
				2226	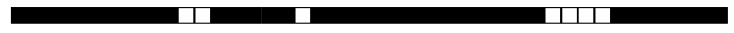	4
				119	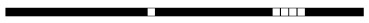	3
				336	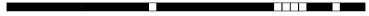	1
		X2	3	18	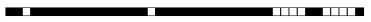	1
				137	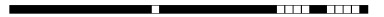	1
				2016	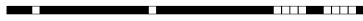	1
		X3	27	92	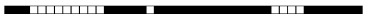	21
				2286	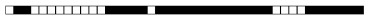	4
				2020	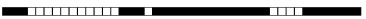	2
T	34	T1	29	53	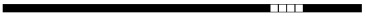	13
				926	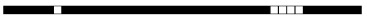	4
				334	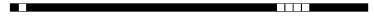	2
				501	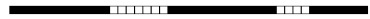	2
				156	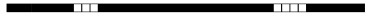	1
				245	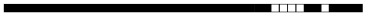	1
				373	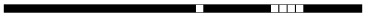	1
				519	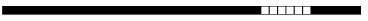	1
				732	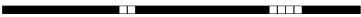	1
				1122	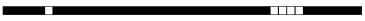	1
				1144	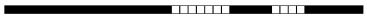	1
				ORPHAN	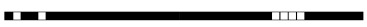	1
		T2	1	52	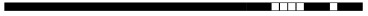	1
		T2/3	2	73	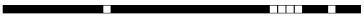	2
		T3	1	ORPHAN	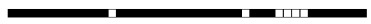	1
		T5-RUS1	1	254	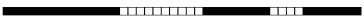	1
		T-TUSCANY	2	1737	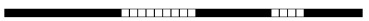	2
S	31			34	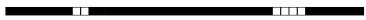	18
				789	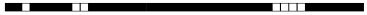	4
				71	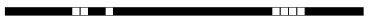	3
				Not in SITVIT	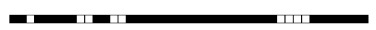	2
				790	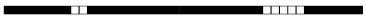	1
				1211	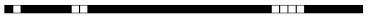	1
				Not in SITVIT	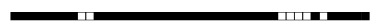	1
				Not in SITVIT	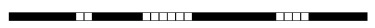	1
EAI	16	EAI1-SOM	10	806	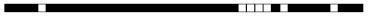	6
				48	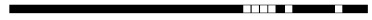	2
				1649	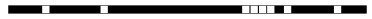	2
		EAI5	5	625	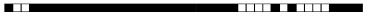	3
				ORPHAN	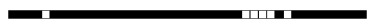	2
		EAI	1	Not in SITVIT	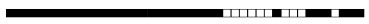	1
H	6	H1	5	62	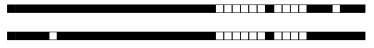	2
				2375	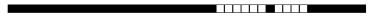	2
				47	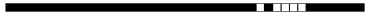	1
		H3	1	50	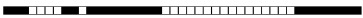	1
CAS	5	CAS1-Kili	2	21	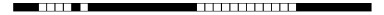	2
		CAS1-Delhi	1	1092	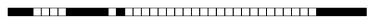	1
		CAS	2	Not in SITVIT	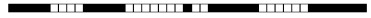	2
unknown	4	Unknown	1	2018	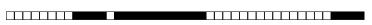	1
		Not in SITVIT	3	Not in SITVIT	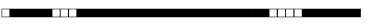	1
				Not in SITVIT	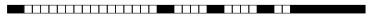	1
				Not in SITVIT	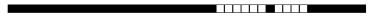 67	1
no result						27
TOTAL						441

SIT = Shared International Type; SITVIT = international spoligotyping database; SITIV 2 = is a genotyping molecular markers database focusing on *Mycobacterium tuberculosis* complex; CAS = Central Asian; Orphan = isolates showed unknown patterns that were not assigned to any known major lineages in the SITVIT2 database; LAM: Latin American; EAI: East-African Indian; Delhi/CAS: Delhi/Central Asian, H: Haarlem); no = number.

**Table 3 pathogens-12-00475-t003:** Distribution of Mtb lineages in this study in comparison with other studies.

		Our Study n (%)	Western Cape n (%) [43]	Gauteng n (%) [43]	KZN n (%) [43]	Free Staten (%) [51]	Limpopo n (%) [44]	North- West n (%) [43]	Zambia n (%) [52]	Botswana n (%) [53]
Lineage	Family	441	897	142	230	86	226	358	274	458
2	Beijing	185 (42)	599 (66.8)	44 (31.0)	57 (24.8)	5 (5.8)	34 (15.0)	88 (24.6)	1 (0.4)	41 (9.0)
4	LAM	83 (18.8)	53 (5.9)	29 (20.4)	42 (18.3)	18 (20.9)	60 (26.5)	54 (15.1)	149 (54.4)	150 (32.8)
4	X	48 (10.9)	88 (9.8)	9 (6.3)	14 (6.1)	5 (5.8)	12 (5.3)	27 (7.5)	19 (6.9)	75 (16.4)
4	T	34 (7.7)	61(6.8)	18 (12.7)	29 (12.6)	14 (16.3)	43 (19.0)	60 (16.8)	39 (14.2)	73 (15.9)
4	S	31 (7.0)	23 (2.6)	9 (6.3)	49 (21.3)	6 (7.0)	21 (9.1)	37 (10.3)	4 (1.5)	62 (13.5)
1	EAI MANU	16 (3.6) 0	6 (0.7) 0	12 (8.5) 3 (2.1)	6 (2.6) 2 (0.9)	0 0	11 (4.9) 3 (1.3)	24 (6.7) 6 (1.7)	6 (2.2)	31 (6.8) 2 (0.4)
4	H	6 (1.4)	10(1.1)	6 (4.8)	8 (3.5)	1 (1.2)	31 (13.7)	26 (7.3)	0	21 (4.6)
3	CAS	5 (1.1)	8 (0.9)	2 (1.4)	5 (2.2)	0	10 (4.4)	6 (1.7)	44 (16.1)	2 (0.4)
3	U	0	7 (0.8)	0	1 (0.4)	0	1 (0.4)	1 (0.3)	0	3 (0.7)

## Data Availability

Data can be requested from the corresponding author.

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
