# Peer review of "Spatial Distribution of Drug-Resistant *Mycobacterium tuberculosis* Infections in Rural Eastern Cape Province of South Africa"

_pathogens, 2023, doi:10.3390/pathogens12030475_

Round 1
Reviewer 1 Report
A very well-written manuscript. A few comments:
1. What was the average distance between the various clinics or healthcare facilities? If the distance was not too much, then how would the difference in mutation rates be explained? This should be discussed in detail.
2. Page 14, line 326: The authors state that Beijing was the only genotype found in hospitals/clinics. This statement is not clear since other genotypes were found too.
Author Response
Dear Sir/Madam
Please see the attached table with responses to Reviewers.
Regards

Reviewer 2 Report
Dear authors,
The manuscript is interesting. However, several critical gaps have not been taken into account:
1. The sample size cannot be chosen randomly.
2. Many of the acronyms are not described in the manuscript.
3. There are multiple writing errors.
4. The figures need to be improved.
5. This manuscript describes the importance of MDR strains. However, this information is not used to emphasize this problem in the samples analyzed. The rpoB, katG, and inhA genes associated with Rifampicin Resistance and Isoniacity have been studied. Moreover, more than one isolate has probably presented modifications in these genes (indicating the presence of MDR strains), and this data was not taken into account.
6. A new analysis of MDR strains needs to be done with the information generated in this manuscript.
7. Why LPA method was selected and not qPCR; what are the advantages or disadvantages of using an LPA for this type of research?
Some clarifications are needed for the following errors:
Figure 1. The map shows the healthcare facilities, but we do not know if the map is from South Africa. Also, not all healthcare facilities are listed in the figure.
Lane 95, which province?
Lane 97. O.R Tambo district, O.R means?
Lane 103, remove the apostrophe.
109 Lane, National Laboratory Services (NHLS), H means?
Lane 121. Sample size should be determined by some specific statistical method; a random analysis is not ideal
152 Lane, USAP means?
Lane 155, BWA, SMALT was not previously described.
Table 1. nil means?
Lane 227: Please, mention all criteria required for “quality control”.
The numbers in Figure 12 are almost invisible.
Author Response
Dear Sir/Madam
Please see the attached table with responses to reviewers.
Regards

Reviewer 3 Report
The manuscript of L.M. Faye and co-authors titled "Spatial distribution and clustering of drug-resistant Mycobacterium tuberculosis infections in rural Eastern Cape Province of South Africa" is devoted to clustering of Mtb infections in South Africa using LPA, spoligotyping and NGS.
The major issue with the manuscript is the lack of publically available NGS data. The authors claimed that they performed NGS for 36 isolates of Mtb (line 129), but neither the platform used nor other details are given in the text. No accession numbers. The description of methods, where it exists, is extremely incomplete.
Other issue include:
Buffalo (line 85) or Bufalo (Figure 1)?
Line 112: How the mutations in the genes were determined?
Line 121: Major issue: How the randomization was performed?
Line 129: How pairwise genetic differences were estimated?
Line 161: How the clustering was performed in R? By using homemade scripts?
Line 174: You give the number of mutations in different HCFs, but the number of isolates in which HCF is not given anywhere. Because of that, the rest figures are useless.
Tables 1 and 2: What means Nil?
Figures 10-14: What means hetero-graph? Where are the axis labels?
Figure 12: Where is color scheme?
Lines 311-314: Ref. 42 is missing in the text.
Table 2: Ref. 59 is missing in the References.
Author Response

(The authors gave the same response as above.)

Round 2
Reviewer 2 Report
Dear authors,
Although the manuscript is very interesting, the present version still maintains the errors previously pointed out:
1. The authors still indicate that the samples were randomly selected
Lanes 126, 127: 441 samples randomly selected?
No sample size criteria are defined in the present manuscript; at least, there is no statistical methodology related to the sample size mentioned in the current manuscript.
2. Please add a figure with the spoligotypes (patterns) results for each resistant sample:
It should be good to see the patterns of IMR-TB, RMR-TB, MDR-TB, heteroresistance, and the wild type.
3. We still have many issues with the abbreviations: Please use only one criterion
Examples of typos observed in the manuscript:
Lane 110: INH monoresistance is abbreviated as (IMR)-TB
Lane 111: RIF resistance as (RMR-TB
Lane 204: RR-TB
Lane 127: INR, IF monoresistance
Lanes 96, 97: O.R Tambo
Lane 99: O R Tambo
Lane 222: OR Tambo
4. When did the authors consider INH monoresistant?
Is it when a mutation on inhA is identified?
Or is it when mutation on katG is identified?
Or both?
Show us a figure with the specific result.
Reviewer 3 Report
The authors have considered and responded to the comments made.
Author Response
The manuscript has been submitted for language editor in MDPI editing service.
Round 3
Reviewer 2 Report
Dear Authors,
The present version shows us a significant improvement.
Please include a footnote in table 2, and describe the meaning and differences between the acronyms: SITVIT, SITVIT2, CAS, Orphan, no and nr.
